# Passive Source Localization Using Compressive Sensing

**DOI:** 10.3390/s19204522

**Published:** 2019-10-17

**Authors:** Hangfang Zhao, M. Jehanzeb Irshad, Huihong Shi, Wen Xu

**Affiliations:** 1College of Information Science & Electronic Engineering, Zhejiang University, Hangzhou 310027, China; hfzhao@zju.edu.cn (H.Z.); jehanzeb.irshad@uog.edu.pk (M.J.I.); shihuihong@yeah.net (H.S.); 2Key Laboratory of Ocean Observation-Imaging Testbed of Zhejiang Province, Zhoushan 316021, China; 3Department of Electrical Engineering, University of Gujrat, Gujrat 50700, Pakistan

**Keywords:** matched-field processing, sparse reconstruction, compressive sensing, source localization, high resolution, sonar, robustness

## Abstract

This paper presents an underwater passive source localization method by forming an underdetermined linear inversion problem. The signal strength on a specified grid is evaluated using sparse reconstruction algorithms by exploiting the spatial sparsity of the source signals. Our strategy leads to a high ratio of measurements to sparsity (RMS), an increase in the peak sharpness with a low side lobe level, and minimization of the dimensionality of the problem due to the formulation of the system equation of the multiple snapshots based on the data correlation matrix. Furthermore, to reduce the computational burden, pre-locating with Bartlett is presented. Our proposed technique can perform close to Bartlet and white noise gain constraint processes in the single-source scenario, but it can give slightly better results while localizing multiple sources. It exhibits the respective characteristics of traditionally used Bartlett and white noise gain constraint methods, such as robustness to environmental/system mismatch and high resolution. Both the simulated and experimental data are processed to demonstrate the effectiveness of the method for underwater source localization.

## 1. Introduction

Matched-field processing (MFP) has been extensively studied for passive source localization [1]. It works by scanning through spatial grids of potential targets while looking for the best matching of the observed data and the sound propagation physical model in an ocean waveguide associated with the scanning target position. In regards to the performance metrics, such as resolution and robustness, existing MFP methods have their own pros and cons [1,2,3]. For example, conventional (Bartlett) MFP is relatively robust, however suffering from some fundamental limitations such as low resolution and a high side lobe level. On the other hand, methods such as Minimum Variance Distortionless Response (MVDR) offer a high resolution and low side lobe level [4,5], however often showing significant sensitivity to environmental/system mismatch; hence, when the assumed and true models differ [6,7], the methods frequently give biased estimates [8].

In many practical situations, the number of source signals is rather small at any time instant compared with the number of scanning grids. On that basis, we can exploit the spatial sparsity feature of the underwater source signals and develop a high-resolution MFP method based on sparse reconstruction algorithms, which have been attracting great interest due to their relationship with Compressive Sensing (CS) [9]. According to the CS theory, the spatial sparsity of a signal can be enforced to enable reconstruction with high probability [10,11,12]. As such, it provides a framework to obtain sparse solutions to underdetermined problems as long as the underlying signal is sparse and the replica dictionary that maps the underlying signal to the observations is sufficiently incoherent. Owing to this ability, CS is widely and successfully used in many applications, e.g., control systems [13,14], magnetic resonance image reconstruction [15,16], computer vision [17], radar detection [18,19,20], geophysics and remote sensing [21,22], speech processing [23], image processing [24,25], error correction in channel coding and estimation [26,27], oceanic engineering [28], pattern recognition and machine learning [29,30], acoustic source localization [11,31,32,33,34], etc.

In this research effort, the solution set inherently is sparse, having substantially fewer sources than possible positions [35]. One of the attractive features of CS is its achievement of super resolution beyond the Rayleigh limit for the single snapshot [36]. However, the performance of CS degrades in the presence of mismatch caused by the misalignment between the actual source field received at the sensor array and the modeled replica vector, and the grid refinement approach can mitigate this problem [37]. In this framework, the passive source localization problem is represented by solving a linear matrix equation, in which the signal strength on a specified grid forms the unknown vector to be solved [38]. Because the grids are often densely specified to obtain a high resolution, the matrix equation is always underdetermined. Because of utilizing the prior knowledge of sparsity, the advantages of the sparse reconstruction-based MFP method can include high resolution, low sidelobe level, robustness, and no need for a large number of snapshots.

Herein, we present a sparse-reconstruction-based MFP method for underwater passive source localization, which builds the measurement equation according to the data correlation matrix; in the body of the paper, we refer to the method as Compressive Sensing-Revised (CS-R). The merits of CS-R include a sharp peak and low side lobe level, the Ratio of Measurements to Sparsity (RMS) gains, and the provision of robust estimation. RMS is defined as the ratio of measurements to sparsity; if we have *M* number of measurements, which only contains a *Q* non-zero value, then the RMS of the signal is M/Q. Furthermore, in order to avoid intractable computations for refining the grid over the whole region, Bartlett MFP is used for pre-locating: sparse-reconstruction-based MFP is applied to the regions where the Bartlett output exceeds the noise threshold [10]. The Bartlett MFP [39] is the most robust method and is extensively used in papers on MFP, and thus, it is an appropriate candidate for a baseline comparison. We consider this process using single and multiple incoherent sources. The White Noise gain Constraint (WNC) process offers potentially high-resolution source localization performance [40]. The WNC is a good benchmark candidate because, unlike MVDR or MUSIC, it is more robust to replica mismatch in realistic scenarios. We consider WNC using single and multiple sources as well. The paper is organized as follows. In Section 2, we describe the passive source localization problem and give a brief overview of several traditional MFP methods. In Section 3, we describe in detail the MFP method in the perspective of sparse representation for a single snapshot. Then, we extend the approach to handle multiple samples in Section 4. Related work is introduced and leading to the CS-R technique. Section 5 investigates the performance of our proposed method versus other methods by means of numerical simulations. Experimental data are processed in Section 6. Finally, some conclusions are drawn in Section 7.

## 2. Theoretical Background

### 2.1. Data Model

We consider a waveguide model for the seismo/acoustic environment in which a point source radiates the narrow-band signal and the signal field is sampled by a receiver array. Some assumptions on signal/noise/environment processes are made [41].
The sources are stationary random processes and uncorrelated with each other. A number of independent snapshots are available.Both the signal and noise processes follow a zero-mean complex Gaussian distribution, and the noise is independent of the signal processes.The source motion effect is ignored, and random environmental variation is stationary.

We assume a network of *M* sensors and *Q* sources located at $r_{q}$, q=1,2,⋯,Q, and these sources emit a narrow-band signal at frequency *f*. Suppose we have *L* independent measurements; the complex envelope of the received signal by *M* number of sensors at the $l^{\mathrm{th}}$ snapshot, $y_{l}\left(f\right)$, can be expressed as [41,42]:
(1)$$y_{l}\left(f\right)=\sum_{q=1}^{Q}{\tilde{b}}_{l,q}\left(f\right)g\left(f,r_{q}\right)+n_{l}\left(f\right),l=1,2,\cdots ,L,$$
where $g\left(f,r_{q}\right)$ is an M×1 vector of Green’s function from the source located at $r_{q}$ to the receiver array; ${\tilde{b}}_{l,q}\left(f\right)$ is a random process of the $q^{\mathrm{th}}$ source incorporating amplitude and phase variability; $n_{l}\left(f\right)$ is a stationary noise vector. For each sample, the received signal can be written in a matrix equation:
(2)$$y_{l}\left(f\right)=G\left(f\right)b_{l}\left(f\right)+n_{l}\left(f\right),$$
where $G\left(f\right)$ is composed of Green’s function corresponding to each source location as its columns: $G\left(f\right)=\left[g\left(f,r_{1}\right),g\left(f,r_{2}\right),\cdots ,g\left(f,r_{Q}\right)\right]$; the source signals group into a vector: $b_{l}\left(f\right)={\left[{\tilde{b}}_{l,1}\left(f\right),{\tilde{b}}_{l,2}\left(f\right),\cdots ,{\tilde{b}}_{l,Q}\left(f\right)\right]}^{T}$. Collection of all the snapshots results in:
(3)Y=GB+N,
where the columns of Y indicate different time samples: $Y=\left[y_{1}\left(f\right),y_{2}\left(f\right),\cdots ,y_{L}\left(f\right)\right]$, and B and N are defined similarly. Thus, the goal of underwater source localization is to determine the columns of matrix G, which is related to the source locations, based on the known measurements Y and unknown signals B.

Several MFP algorithms either explicitly or implicitly involve the formation of the sample covariance matrix ${\hat{K}}_{Y}$, given by:
(4)$${\hat{K}}_{Y}=\frac{1}{L}\sum_{l=1}^{L}y_{l}\left(f\right)y_{l}^{H}\left(f\right).$$

Some adaptive MFP algorithms need enough snapshots to make ${\hat{K}}_{Y}$ full rank. However, in the time-varying channel and high-speed source cases, the number of snapshots available for accumulating ${\hat{K}}_{Y}$ is limited [6,7]. Thus, diagonal loading can be applied, and the sample covariance matrix is replaced by:
(5)$${\hat{K}}_{Y}=\frac{1}{L}\sum_{l=1}^{L}y_{l}\left(f\right)y_{l}^{H}\left(f\right)+\mu I,$$
where μ is a small constant [1].

### 2.2. The Traditional MFP Methods

The Bartlett MFP correlates the data vector with the replica vector and scans all the search region to find the maximum output [1]. The output of Bartlett MFP can be written as:
(6)$$B_{Bart}\left(r\right)=\frac{1}{L}\sum_{l=1}^{L}{\left|w_{c}^{H}\left(r\right)y_{l}\right|}^{2}=w_{c}^{H}\left(r\right){\hat{K}}_{Y}w_{c}\left(r\right),$$
where the replica vector $w_{c}\left(r\right)$ is the normalized Green’s function vector,
(7)$$w_{c}\left(r\right)=\frac{g\left(f,r\right)}{{∥g\left(f,r\right)∥}_{2}}.$$

It is well known that Bartlett MFP is relatively easy to implement, ant-interference, and not sensitive to environmental mismatch, but it suffers from a broad main lobe and high sidelobe level, which results in low resolution. The narrow main lobe and sidelobe suppression are important if more than one source is present. Later, we use Bartlett MFP as the pre-treatment of other methods.

The WNC method is a high-resolution method, as it suppresses sidelobes while offering a degree of robustness in frequently-encountered system/environmental mismatch scenarios. The WNC is a potential standard as, unlike MVDR, it is more robust to system/environmental mismatch [1,4]. Its output is given by:
(8)$$B_{WNC}\left(r\right)=\left[w_{n}^{H}\left(r\right){\hat{K}}_{Y}w_{n}\left(r\right)\right],$$
where:$$w_{n}\left(r\right)=\frac{{\left({\hat{K}}_{Y}+\gamma I\right)}^{-1}w_{c}\left(r\right)}{w_{c}^{H}\left(r\right){\left({\hat{K}}_{Y}+\gamma I\right)}^{-1}w_{c}\left(r\right)},$$

The adaptive weights $w_{n}$ correspond to diagonally-loaded MVDR weights and are obtained by solving:
(9)$$\begin{array}{cc} \min_{w_{n}}\;w_{n}^{H}\left(r\right){\hat{K}}_{Y} & w_{n}\left(r\right)\;\mathrm{subject}\;\mathrm{to}\\ w_{n}^{H}w_{c}\left(r\right) & =1,\\ {\left|w_{n}^{H}w_{n}\right|}^{-1} & \geq \kappa^{2}. \end{array}$$

To satisfy a white noise gain constraint $G_{WNG}$, this iterative algorithm finds the diagonal loading γ for each replica vector such that:
(10)$$\kappa^{2}<G_{WNG}={\left|w_{n}^{H}w_{n}\right|}^{-1}<M.$$

In practice, we normalize it and express it as 10 $\log_{10}\left(\kappa^{2}/M\right)\leq 0$ dB [43]. The WNC process is a better choice for baseline comparison because it has been shown in many articles that the MVDR has a better resolution than Bartlett MFP, but it is sensitive to environmental/system mismatch and needs enough snapshots to make ${\hat{K}}_{Y}$ full rank [2,3,5].

## 3. Sparse Reconstruction-Based MFP with a Single Snapshot

In this section, we start to formulate the MFP problem as a sparse representation problem. For simplicity, we describe the sparse reconstruction-based MFP technique for a single time sample in this section, then the approach will be extended to the multiple snapshot case in the next section.

Instead of estimating the source locations directly, we discretize the region containing the sources with an *N*-point grid, $\left\{r_{1},r_{2},\cdots ,r_{N}\right\}$, and assume that each source is located on one of the grid points. The signal strength on each grid point forms the unknown vector $b_{CS}$ to be solved, $b_{CS}={\left[{\tilde{b}}_{1},{\tilde{b}}_{2},\cdots ,{\tilde{b}}_{N}\right]}^{T}$, where the $n^{\mathrm{th}}$ element ${\tilde{b}}_{n}$ is nonzero if some source comes from $r_{n}$ and zero otherwise. In practice, there are usually very few sources within the interested area, with the number much smaller than the grid points. Thus, we have reached a sparse representation of the spatial distribution of sources $b_{CS}$.

We construct a replica matrix based on the *N*-point grid,
(11)$$G_{CS}=\left[g\left(f,r_{1}\right),g\left(f,r_{2}\right),\cdots ,g\left(f,r_{N}\right)\right],$$
where $g\left(f,r_{n}\right)$ is Green’s function from a test source located at $r_{n}$ from the receiver array. Since the number of grid points is always larger than the measurements (that is N≫M), $G_{CS}$ is overcomplete. In this framework, the measurement matrix is known and does not depend on the actual source locations.

For a single time sample, we reach the measuring equation:
(12)$$y=G_{CS}b_{CS}+n.$$

Different from (Equation 2), the columns of $G_{CS}$ and the rows of $b_{CS}$ are indexed by grid positions $\left\{r_{1},r_{2},\cdots ,r_{N}\right\}$. Therefore, the MFP problem is translated into a sparse signal reconstruction problem, which is to determine the source strength on each grid point $b_{CS}$ based on the measurements y and known measurement matrix $G_{CS}$.

In this case, Equation (Equation 12) is underdetermined and does not have a unique solution. Recent developments in the CS theory have shown that a *Q*-sparse *N*-dimensional signal (only has *Q* non-zeros entries) can be uniquely reconstructed by $O\left(Q\log\left(N/Q\;\right)\right)$ incoherent linear measurements [44]. Therefore, Equation (Equation 12) can be solved through sparse reconstruction algorithms, such as Basis Pursuit (BP) [45,46,47], Matching Pursuit (MP) [45,48], and Compressive Sampling Matching Pursuit (CoSaMP) [49]. In this paper, we adopt the BP algorithm, which has been shown to be near optimal and robust to noise and to coherent dictionary entries as well [43]. Thus, in the perspective of spatial sparse constraint, the MFP problem is formulated as:
(13)$$\min{∥b_{CS}∥}_{1},s.t.{∥y-G_{CS}b_{CS}∥}_{2}^{2}\leq \varepsilon^{2}.$$

The regularization parameter ε is introduced to tolerate or penalize the noise in the measurements. ε should be set appropriately: if ε is too small, it will not be able to tolerate noise, causing several peaks in the output; if ε is too large, there will be no penalty for noise, always resulting in a zero solution. In (Equation 13), the $l_{1}$-norm is imposed on $b_{CS}$ to emphasize sparsity in space; thus, the sparse-reconstruction-based MFP can achieve high resolution and suppress the side lobes effectively. RMS can be set as a performance indicator of sparse reconstruction-based MFP methods. The higher the RMS, the better the performance of sparse reconstruction [50,51,52]. The RMS of (Equation 12) is M/Q (*M* measurements for *Q* sparsity).

## 4. Sparse Reconstruction-Based MFP with Multiple Snapshots

For multiple snapshots, we can reach:
(14)$$Y=G_{CS}B_{CS}+N,$$
where $B_{CS}=\left[b_{CS,1},b_{CS,2},\cdots ,b_{CS,L}\right]$. The matrix $B_{CS}$ is parameterized temporally and spatially, but it is only sparse in space, with QL non-zeros. The numerical solution of (Equation 14) is a bit more involved than that of a single sample case, since we should solve the problem in the N×L-dimensional complex space.

### 4.1. Related Work

Malioutov et al. provided some ideas on solving (Equation 14) and reducing the problem’s dimensionality [11]. The first thought is separate treatment of each snapshot, thus reducing the problem dimensionality to *N*, and finally averaging the results across all the snapshots. This approach is simple, but it has low performance and is sensitive to noise. The second approach is to treat different time samples in synergy. Since matrix $B_{CS}$ is only sparse in space, one can compute the $l_{2}$-norm of each row of $B_{CS}$, given by $b^{\left(l_{2}\right)}$, which is a *Q*-sparse *N*-dimensional signal. Penalize the $l_{1}$-norm of $b^{\left(l_{2}\right)}$, and the cost function becomes:
(15)$$\min{∥b^{\left(l_{2}\right)}∥}_{1},s.t.{∥Y-G_{CS}B_{CS}∥}_{2}^{2}\leq \varepsilon^{2}.$$

This treatment can achieve better performance than treating each snapshot separately, but the computational cost for BP is $O\left({\left(N\times L\right)}^{3}\right)$, which is greatly increased by the number of snapshots. Malioutov et al. proposed a Singular-Value Decomposition (SVD) method to reduce the dimensionality. They took the SVD of the measurement data matrix $Y=U\Sigma V^{H}$ and kept the components in signal subspace, which resulted in an M×Q matrix $Y_{SV}=YV_{Q}$, where $V_{Q}$ is the first *Q* columns of V corresponding to the *Q* largest singular values. Thus, the measurement equation is transformed to:
(16)$$Y_{SV}=G_{CS}B_{SV}+N_{SV},$$
where $B_{SV}=BV_{Q}$ and $N_{SV}=NV_{Q}$. The problem dimensionality is reduced to N×Q; however, the RMS is also retained as M/Q (M×Q measurements for $Q^{2}$ sparsity), and it requires sources to be sufficiently separated. We consider this method to be closely related to our work. It is also a good candidate for the comparison with our proposed method. We call this the 1-SVD method in the rest of this paper.

Mantzel et al. also proposed CS imaging for MFP named as cMFP, where it focuses more on the computational complexity reduction rather than the imaging model and recovery performance [31]. The method proposed in our work mainly focuses on performance and partially on the reduction of the computational burden. We draw a comparison of this method with our proposed method in terms of performance, and the results show that our method has clear advantages in terms of the localization of the source and main lobe to sidelobe ratio.
(17)$$\arg\min{∥\Phi (Y-G_{CS}B_{CS})∥}^{2}.$$
where Φ is an M×N encoding matrix. The question remains as to how to choose the encoding matrix Φ so that the solution to this method is similar to the standard MFP, e.g., Bartlett. To reduce the computational burden, the construction of Φ is proposed to be a random linear mapping. The performance of the cMFP will match that of the standard MFP when Φ maintains the energy of the differences between the observations Y and all scalar multiples of Green’s function at different points. This method is used for comparison, and it is named cMFP in this paper.

### 4.2. CS-R Method

In this section, we present a new approach, which can extend the use of sparse signal representation ideas to practical underwater source localization problems. It is convenient to write the data correlation matrix of (Equation 14) as:
(18)$${\hat{K}}_{Y}=YY^{H}/L=G_{CS}{\hat{K}}_{B}G_{CS}^{H}+{\hat{K}}_{N}+{\hat{K}}_{GBN}+{\hat{K}}_{NGB},$$
where ${\hat{K}}_{B}=B_{CS}B_{CS}^{H}/L$, ${\hat{K}}_{N}=NN^{H}/L$, ${\hat{K}}_{GBN}=G_{CS}B_{CS}N^{H}/L$, ${\hat{K}}_{NGB}={\hat{K}}_{GBN}^{H}$. In the case of white noise, there is ${\tilde{K}}_{N}={\hat{K}}_{N}-\sigma_{n}^{2}I$ (where $\sigma_{n}^{2}$ is the noise variance), and hence, all the entries in ${\tilde{K}}_{N}$, ${\hat{K}}_{GBN}$ and ${\hat{K}}_{NGB}$ are zero mean. Since the sources are uncorrelated with each other, ${\hat{K}}_{B}$ only has large entries along its diagonal. Therefore, Equation (Equation 18) can be approximated as:
(19)z=Ad+e,
where:
(20)$$\begin{array}{cc} z & =vec\left({\hat{K}}_{Y}-\sigma_{n}^{2}I\right),\\ A & =\left[vec\left(g\left(r_{1}\right)g^{H}\left(r_{1}\right)\right),vec\left(g\left(r_{2}\right)g^{H}\left(r_{2}\right)\right),\cdots ,vec\left(g\left(r_{N}\right)g^{H}\left(r_{N}\right)\right)\right],\\ d & ={\left[{\hat{K}}_{B}\left(1,1\right),{\hat{K}}_{B}\left(2,2\right),\cdots ,{\hat{K}}_{B}\left(N,N\right)\right]}^{T},\\ e & =vec\left({\tilde{K}}_{N}+{\hat{K}}_{GBN}+{\hat{K}}_{NGB}\right), \end{array}$$
where $vec\left(\cdot \right)$ denotes the vectorization operator (stacking the columns of a matrix on top of each other). It is obvious that d is an *N*-dimensional real vector with *Q* sparsity. Since the number of measurements is increased, while the sparsity is retained at the minimum *Q*, the RMS of (Equation 19) is increased.

Since the rows of the measurement matrix A may be linearly dependent and the dimensionality of A can be quite high, SVD is used to reduce the measurement space and increase the SNR. Taking the SVD of A, $A=U\Sigma V^{H}$, reserving the $\bar{M}$ largest singular values $\lambda_{m}$, $m=1,2,\cdots ,\bar{M}$, and projecting the measurements onto the corresponding output basis vectors, we derive the following equation:
(21)$$\bar{z}={\bar{U}}^{H}z=\bar{\Sigma }{\bar{V}}^{H}d+{\bar{U}}^{H}e=\bar{A}d+\bar{e},$$
where $\bar{\Sigma }$ is the diagonal matrix consisting of the $\bar{M}$ largest singular values and $\bar{U}$, $\bar{V}$ denote the corresponding $\bar{M}$ columns of U and V, respectively. Generally, after the processing of (Equation 21), almost all the signal power is retained, whereas there is only a part of the noise power left. Thus, SNR increases, which will benefit the latter sparse reconstruction process. Furthermore, the problem dimensionality is decreased to $\bar{M}\times Q$. Equation (Equation 21) can be solved through:
(22)$$\min{∥d∥}_{1},s.t.{∥\bar{z}-\bar{A}d∥}_{2}^{2}\leq \varepsilon^{2}.$$

The schematic diagram of implementing the CS-R method is given in Figure 1.

### 4.3. Restricted Isometry Property for Sparse Recovery

The Restricted Isometry Property (RIP) for sparse recovery is related to the orthonormality of the columns of the A matrix. The RIP condition is a necessary, though not sufficient, condition for signal recovery [46]. Let the sparsity *Q* of a vector be the number of nonzero elements, and define the RIP constant $\delta_{Q}<1$ to be the smallest positive number such that the inequality:
(23)$$\left(1-\delta_{Q}\right){∥d∥}_{2}^{2}\leq {∥\mathrm{Ad}∥}_{2}^{2}\leq \left(1+\delta_{Q}\right){∥d∥}_{2}^{2}$$
for every vector d which is *Q*-sparse. Here, we can define another quantity, the $Q,\overset{´}{Q}$-restricted orthogonality constant $\theta_{Q,\overset{´}{Q}}$, as the smallest number that satisfies:
(24)$$|\langle \mathrm{Ad},A\overset{´}{d}\rangle |\leq \theta_{Q,\overset{´}{Q}}{∥d∥}_{2}∥\overset{´}{d}{∥}_{2}$$
for all d and $\overset{´}{d}$ such that d and $\overset{´}{d}$ are *Q*-sparse and $\overset{´}{Q}$-sparse, respectively, and have disjoint supports. The constants $\delta_{Q}$ and $\theta_{Q,\overset{´}{Q}}$ are related by the following inequalities,
(25)$$\theta_{Q,\overset{´}{Q}}\leq \delta_{Q,\overset{´}{Q}}\leq \theta_{Q,\overset{´}{Q}}+\max\left(\delta_{Q},\delta_{\overset{´}{Q}}\right)$$

A sensing matrix is said to satisfy the 2Q-RIP if $\delta_{2Q}<\sqrt{2}-1$ [52]. In this work, the 2Q-RIP condition is always satisfied for a noiseless environment, where the sparse target is Q=1. The condition in the presence of bounded error is also fulfilled; the value of 0.98 is incurred, where that condition is given as $\delta_{2Q}+\theta_{Q,2Q}<1$ [53]. A direct consequence of (Equation 23) is that $\delta_{2Q}<\sqrt{2}-1$ is in fact a strictly stronger condition than $\delta_{2Q}+\theta_{Q,2Q}<1$ [54].

### 4.4. Comparison between the CS-R Method and the Bartlett Method

The output of Bartlett is $B_{Bart}\left(r\right)=w_{c}^{H}\left(r\right){\hat{K}}_{Y}w_{c}\left(r\right)$, where $w_{c}\left(r\right)=\frac{g\left(f,r\right)}{{∥g\left(f,r\right)∥}_{2}}$. For the system equation of CS-R (Equation 18), ignoring the noise ${\hat{K}}_{N}+{\hat{K}}_{GBN}+{\hat{K}}_{NGB}$, we can simplify the system equation as ${\hat{K}}_{Y}=G_{CS}{\hat{K}}_{B}G_{CS}^{H}$. Then, ${\hat{K}}_{Y}$ can be expressed as:
(26)$${\hat{K}}_{Y}=\left(G_{CS}Q^{-1}\right)Q{\hat{K}}_{B}Q^{H}{\left(G_{CS}Q^{-1}\right)}^{H},$$
where $Q=diag\left({∥g\left(r_{1}\right)∥}_{2},{∥g\left(r_{2}\right)∥}_{2},\cdots ,{∥g\left(r_{N}\right)∥}_{2}\right)$. $G_{CS}Q^{-1}$ denotes normalizing the Green’s function of each grid point; the $n^{\mathrm{th}}$ column of $G_{CS}Q^{-1}$ is $\frac{g\left(f,r_{n}\right)}{{∥g\left(f,r_{n}\right)∥}_{2}}$. Through Least-Squares (LS), we derive the following equations:
(27)$$Q{\hat{K}}_{B}Q^{H}=F{\left(G_{CS}Q^{-1}\right)}^{H}{\hat{K}}_{Y}\left(G_{CS}Q^{-1}\right)F^{H}$$
where $F={\left[{\left(G_{CS}Q^{-1}\right)}^{H}\left(G_{CS}Q^{-1}\right)\right]}^{-1}$. Furthermore, we obtain:
(28)$$F^{-1}Q{\hat{K}}_{B}Q^{H}F^{-H}={\left(G_{CS}Q^{-1}\right)}^{H}{\hat{K}}_{Y}\left(G_{CS}Q^{-1}\right),$$
and:
(29)$${\hat{K}}_{B}\propto {\left(G_{CS}Q^{-1}\right)}^{H}{\hat{K}}_{Y}\left(G_{CS}Q^{-1}\right),$$
where ${\left(G_{CS}Q^{-1}\right)}^{H}{\hat{K}}_{Y}\left(G_{CS}Q^{-1}\right)$ is the objective equation of Bartlett MFP. CS-R translates the system equation into solving (Equation 22), wherein the $l_{1}$-norm is used for the penalty of spatial sparsity, ensuring that CS-R can achieve high resolution and suppress side lobes effectively; the system equation of the $l_{2}$-norm can be seen as equivalent to the objective equation of Bartlett MFP, meaning that CS-R can provide robust estimation similar to that of Bartlett.

Compared to the other sparse-reconstruction-based MFP methods, CS-R has three added advantages: the problem dimensionality is reduced to the minimum, and the unknown signal is real, so the computational cost is lower; there are high RMS gains, with better reconstruction performance; robustness, which is rather important for underwater source localization.

### 4.5. Pre-Locating

Since the MFP problem in our framework is performed on a grid, the accuracy depends on the grid size. However, it is computationally expensive to make the grid uniformly fine on the whole region. Malioutov et al. introduced the idea of adaptive grid refinement, in which the grid is refined around the source locations derived by the previous estimation [11]. Inspired by this, if we have obtained approximate knowledge of source locations using low resolution, but simple and robust methods, such as Bartlett MFP, the grid for sparse reconstruction-based MFP can be specified to cover only the regions where sources are located with a high probability, as shown in Figure 2 [10]. Multiple sub-regions may be selected for multiple targets’ localization. These regions can be treated as a whole, reconstructing the multi-targets at the same time; whereas the isolated regions can also be treated separately, largely reducing the computational load of sparse reconstruction algorithms. Pre-locating has some limitations while locating multiple sources, especially closely located sources or detecting a weak submerged target in the presence of a loud surface interferer. We observe that pre-locating works fine in the single-source scenario; hence, on a prior knowledge basis, we can skip pre-locating in the multi-source scenario.

## 5. Simulation Results

Similar to Bartlett, all those sparse reconstruction-based MFP methods do not require the sample covariance matrix to be full rank; thus, they also do not need a large number of snapshots. Our simulation and experimental data processing will focus on the snapshot deficient case, that is L<M. In this section, we will use numerical simulation to verify that the CS-R method can give results similar to other existing adaptive processes, namely Bartlett, WNC, and 1-SVD. Firstly, we compare the ambiguity surface of CS-R to Bartlett MFP. Next, the source location estimation is performed for CS-R in comparison with Bartlett, WNC, and 1-SVD, followed by the performance analysis with environmental uncertainty.

The simulation scenario is shown in Figure 3. The water depth is 102 m, and the acoustic field is received at an array of M=40 sensors spanning the water column. The source, located at the depth and range of 20 m and 12 km, respectively, emits a narrow-band signal at a frequency of 630 Hz. We further assumed that twenty independent snapshots are available for each estimate (L=20), and SNR = 5 dB. The location of the peak on each ambiguity surface is shown by a circle in the simulation results.

### 5.1. CS-R vs. Bartlett

The results of Bartlett MFP and CS-R are shown in Figure 4, wherein CS-R can trade off between the $l_{1}$-norm and $l_{2}$-norm through the regularization parameter ε. The top row in Figure 4 shows that the Bartlett MFP has low resolution, and its side lobe level is rather high. As shown in Figure 4e,f, a good choice of ε is two; the ambiguity surface of CS-R has a sharp peak, and the peak corresponds to the source location. When ε is 0.3 times 2, many peaks appear, shown in the Figure 4c,d; when ε is set at a higher value, 3 times 2 for example, CS-R degrades to the Bartlett MFP, and its side lobe level grows and main lobe broadens as shown in Figure 4g,h. Besides, the ambiguity surfaces of Bartlett and CS-R have the same multi-modal structure. The example illustrates two points. First, the choice of the parameter ε is very important; when the parameter ε is too large, CS-R degrades into Bartlett MFP, which is because the $l_{2}$-norm of CS-R can be equivalent to Bartlett. In this case, an optimum value of the parameter ε is estimated to achieve a trade-off between high resolution and robustness.

The choice of the regularization parameter ε plays an important role in our source localization framework, which balances the fit of the solution to the data versus the sparsity prior. This question arises in many practical inverse problems, and the parameter selection can be made if the noise statistics are known or can be estimated. The idea is to select ε for the residuals of the solution $\hat{d}\left(\varepsilon \right)$ for some known statistics of the noise when such statistics are available. If the distribution of the noise is known or can be estimated [55], then the regularization parameter can be calculated such that ${∥\bar{z}-\bar{A}\hat{d}\left(\varepsilon \right)∥}_{f}^{2}\approx E{∥e∥}_{f}^{2}$. The Frobenius norm is defined as ${∥e∥}_{f}^{2}={∥vec(e)∥}_{2}^{2}$. When the noise statistics are not known and no knowledge of the number of sources is available, the choice of the regularization parameter is still an open problem.

### 5.2. Source Location Estimation Comparison in a Complex Environment

This simulation is performed using the environment described in Figure 3 for both depth and range estimation of the underwater source location estimation. The results presented in Figure 5 indicate that the Bartlett surface does not have a prominent peak, but provides the right peak position. WNC and 1-SVD provide the right peak position with a sharper surface as compared to Bartlett. cMFP performs closely to Bartlett using random received vector from the source, but a slightly broader main lobe than Bartlett. On the other hand, CS-R achieves a higher resolution than the WNC and 1-SVD with the correct peak position and low side lobe level, which indicates its robustness and high-resolution capabilities. Using the same simulation environment, two sources are localized in Figure 6.

In Figure 6a, the locations of two sources, which have the same depth and are 0.4 km apart in range, are estimated; while Source 1 has a higher SNR (30 dB) and Source 2 exhibits weaker SNR (8 dB). It is observed that Bartlett and cMFP show many peaks higher than the Source 2 level. On the other hand, all other processes find the right peak location, and CS-R has lower sidelobes. Note that pre-locating is not implemented in the multi-source scenario for CS-R. It can be seen in Figure 6b that all processes suffer from a high side-lobe level when there is a 1% mismatch. Bartlett and cMFP again show many false peak locations other than sources. The WNC and CS-R processes perform better than 1-SVD.

### 5.3. Performance Analysis with Environmental Mismatch

Performance analysis using a simulated environmental model can provide important guidance for practical applications. We focus on performance analysis with environmental uncertainty here. While the respective true sound speeds in the sediment are 1579 m/s on the top and 1629 m/s on the bottom, we assume they are 1595 m/s on the top and 1645 m/s on the bottom instead to create a case of some environmental mismatch deliberately. Performance analysis in source range estimation is evaluated to compare Bartlett, WNC, 1-SVD, CS-R, and cMFP. Results are shown in Figure 7, which are derived based on 5000 Monte Carlo simulations with 20 snapshots for every run.
In Figure 7a, we present plots of the the Mean Square Error (MSE) of each method versus SNR. The search for the source range location is between 5 and 15 km, and the depth is between 1 and 102 m.Figure 7b gives the evaluation results of the probability of localization error (Pe) versus SNR. The search for range is between 5 and 15 km, and the depth is between 1 and 102 m.

It is found in Figure 7 that the performance of CS-R and cMFP is much closer to Bartlett than either WNC and 1-SVD, following closely Bartlett in the whole SNR the environmental mismatch scenario for the single source, showing robustness similar to the Bartlett process in the single-source scenario.

The performance comparison for source localization is investigated with environmental mismatch while having two sources, and the results are shown in Figure 8. Results of Bartlett, cMFP, and 1-SVD of this investigation were relatively poor; Bartlett and cMFP were unable to estimate the correct location of the second source due to its broad main lobe and high sidelobe level, and 1-SVD’s performance was similar to Bartlett due to its high sensitivity to mismatch. Figure 8b also compares the probability of error for the multi-source scenario versus SNR. The WNC and CS-R processes performed much better, e.g., at SNR = 5 dB, the probability of the error of the localization of second source was very low for WNC and CS-R. The performance of CS-R was slightly better in the two-source scenario than WNC, as shown in Figure 8.

### 5.4. Peak to Background Ratio Analysis

To demonstrate the high-resolution capabilities of CS-R in underwater source location estimation, which include a sharp main lobe level and a low side lobe level, the Peak to Background Ratio (PBR) analysis was performed.

PBR of CS-R was compared with WNC, Bartlett, cMFP, and 1-SVD while changing different parameters, as shown in Figure 9. The obtained results showed the high resolution advantage of CS-R over other conventional methods in different challenging scenarios, e.g., snapshot deficient and low SNR. Although all processes can perform in the snapshot-deficient scenario, the purpose here is to show that, unlike some other methods, e.g., MVDR, our proposed method can also perform in those kinds of scenarios. We used 20 snapshots in Figure 9a and 20 dB in Figure 9b. We observe in Figure 9 that CS-R achieves PBR better than Bartlett and WNC and is close to 1-SVD. CS-R only struggled when the SNR was low, as shown in Figure 9a, but still exhibited slightly better results than Bartlett and WNC. Equation (Equation 30) below is used to calculate the PBR of different source location estimation methods [56].
(30)$$PBR=\frac{P_{A}}{B_{A}}$$
where $P_{A}$ is the total integrated area within peak boundaries minus the background area and $B_{A}$ is the rectangular area of background height times the length between peak boundaries. Note that the background height is defined as the minimum intensity at peak boundaries, and peak height is defined as the maximum intensity between peak boundaries minus background height.

## 6. Experimental Results

The Asian Seas International Acoustics Experiments (ASIAEX) were carried out in the spring of 2001 in the East China Sea (ECS) [57]. Here, we used part of the data from those experiments for our analysis. The site location is shown in Figure 10. The scenario for the shallow water experiment is shown in Figure 11. The water depth was about 102 m, and the acoustic field was received at a vertical line array consisting of 60 sensors. The source signal frequency was 630 Hz. Conductance Temperature Depth (CTD) sensors were used to calibrate the depth of the acoustic sensors and measure the Sound Speed Profile (SSP) of the water column. The SNR was about 15 dB. The range (determined by GPS coordinates) and depth (prior knowledge) of the source at the time of experiment (see Figure 12 at about 15 hours of time) was about 11.6 km and 40 m, respectively.

After removing noisy channels, we selected 42 sensors and 30 independent snapshots in utilizing experimental data; thus, the dimensionality of the received signal Y was 42×30, which is a snapshot-deficient scenario in the statisticalsense. The search interval for the source location was between 11 and 13 km and 1 and 102 m for range and depth, respectively. Pre-locating was implemented for CS-R, and the discretization of the search grid for CS-R remained the same for all other processes. Figure 13 summarizes the ambiguity surfaces of the Bartlett, WNC, 1-SVD, cMFP, and CS-R processes, respectively.

We observe in Figure 13 that the main lobe of the Bartlett process is broad, and the peak to side lobe ratio is rather low. However, it is most likely to provide an optimum peak location. The cMFP’s results show that it focuses more on the computational complexity reduction rather than the imaging model and recovery performance; therefore, it suffers from a broad main lobe and high side lobe level. The WNC provides a much sharper peak closer to the true source location, but 1-SVD fails to provide the right peak location due to its sensitivity to environmental/system mismatch. It can be seen that the CS-R achieves a slightly sharper spectrum than WNC, but it also has the same peak corresponding to the source location as Bartlett and WNC.

To demonstrate the properties of our proposed method, we constructed the multi-source scenario using real data. A time-delayed copy of the ASIAEX was added to the original ASIAEX to create a distance of about 700 m between two sources. In Figure 14, the results of two-source localization using experimental data are shown. The first source was located at a range of about 11.6 km and a depth of about 40 m, while the second source was located at a range of about 12.3 km and a depth of also about 40 m. Thirty time snapshots from the vertical line array of 42 sensors were used. The SNR for the first and second sources were 15 dB and 8 dB, respectively. The two source results demonstrate that WNC and CS-R performed closely to each other, and CS-R can achieved slightly better results, as shown in the Figure 14. The Bartlett and cMFP processes were unable to show the weaker source’s true location due to the high sidelobe level. On the other hand, 1-SVD remained unable to find the true locations of both sources. In future, the reported technique can be used for different scenarios, i.e., passive detection and tracking of an underwater acoustic target [58].

## 7. Conclusions

In this paper, we considered the underwater source localization problems and explored a high-resolution and robust CS-R method based on sparse reconstruction algorithms for the matched field processing application. In our framework, the source localization problem was formulated to solve an underdetermined equation by discretizing the source space into grids. Then, in order to handle the multiple snapshots, to reduce the signal space dimensionality, and to increase RMS, we established the measurement equation according to the data correlation matrix. CS-R uses the $l_{1}$-norm to penalize the spatial sparsity and the $l_{2}$-norm to penalize the error of the measurement equation. Pre-locating with the Bartlett processes was also proposed to reduce the computational load for the single-source scenario. The results of the numerical simulations suggested that the CS-R method allowed achieving a trade-off between high resolution and robustness through the choice of the regularization parameter. The experimental data were used to show that the performance characteristics of our proposed technique were similar to the adaptive process. CS-R exhibited high resolution capabilities and robustness close to both WNC and Bartlett and showed better performance than 1-SVD and cMFP. Both the single source and multi-sources scenario were used to investigate the performance of different processes. In the single-source scenario, the Bartlett process was more robust, and CS-R could achieve results close to it, but at the cost of increased computational load. In the two-source scenario, the CS-R’s performance was slightly better than WNC. As future work, the performance of CS-R in data replica mismatch can be improved.

## Figures and Tables

**Figure 1 sensors-19-04522-f001:**
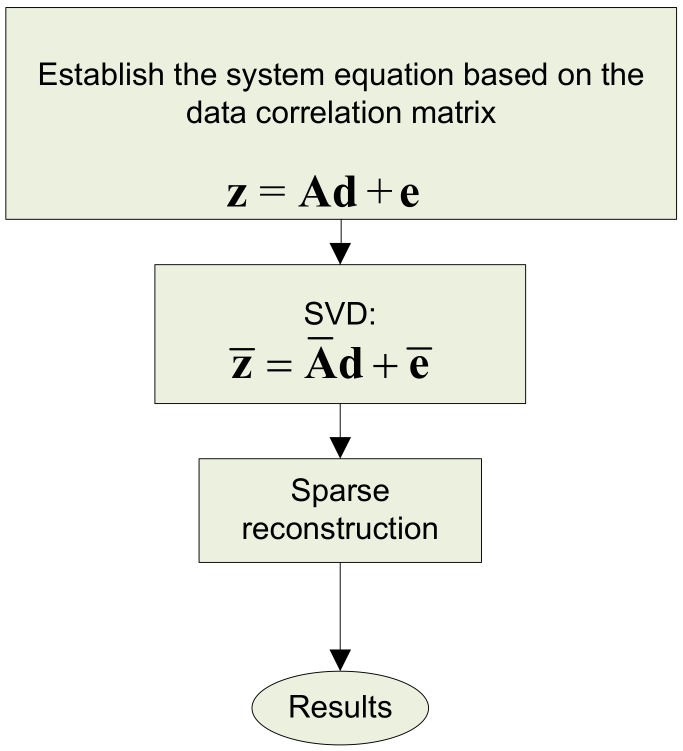
The schematic diagram of implementing the Compressive Sensing-Revised (CS-R) method.

**Figure 2 sensors-19-04522-f002:**
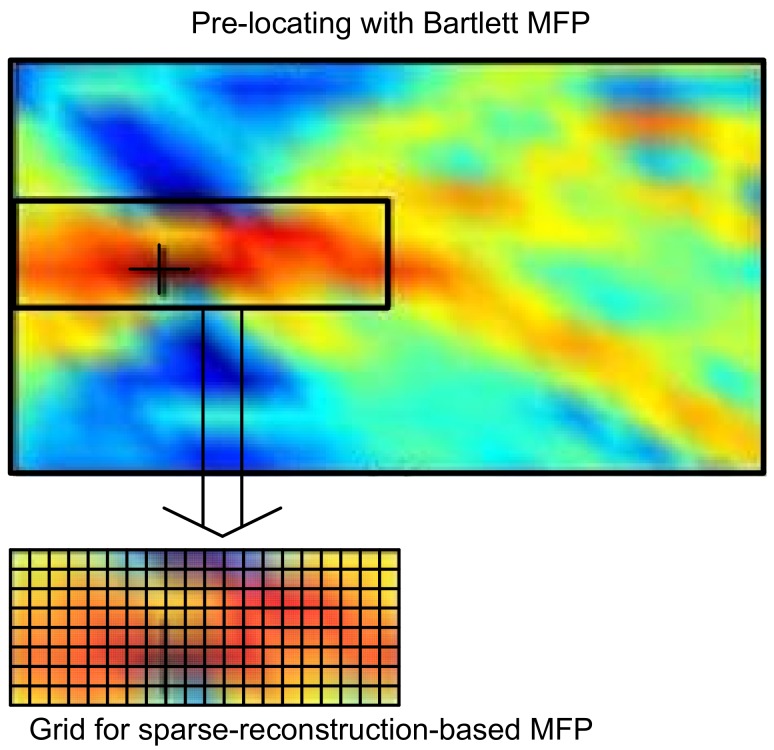
Implementation of pre-locating. MFP, matched-field processing.

**Figure 3 sensors-19-04522-f003:**
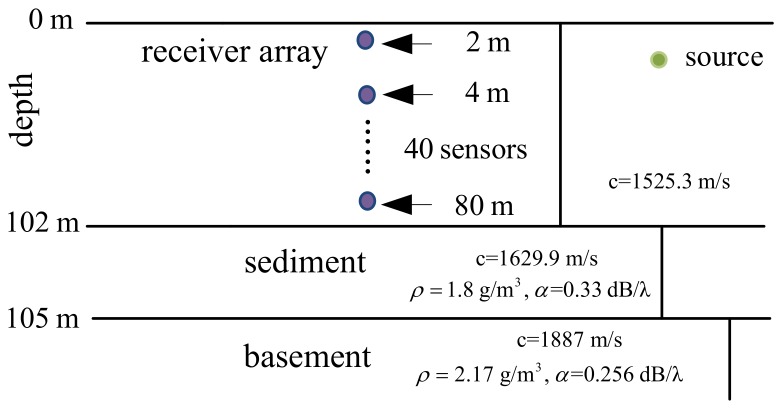
A waveguide environment.

**Figure 4 sensors-19-04522-f004:**
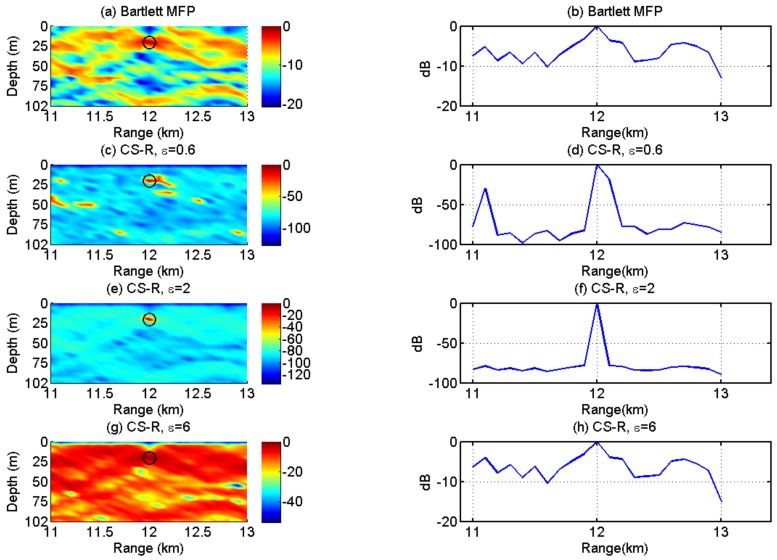
Comparison of Bartlett and CS-R with different choices of parameter ε.

**Figure 5 sensors-19-04522-f005:**
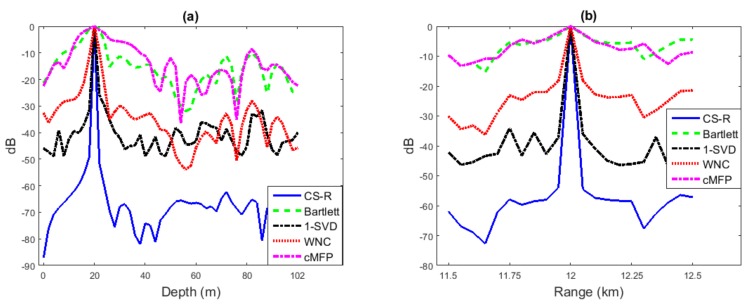
Location estimation from 20 snapshots for a source at depth of 20 m and a range of 12 km with a uniform vertical line array with M=40 sensors and with SNR = 30 dB: (**a**) for depth and (**b**) for range. WNC, White Noise gain Constraint.

**Figure 6 sensors-19-04522-f006:**
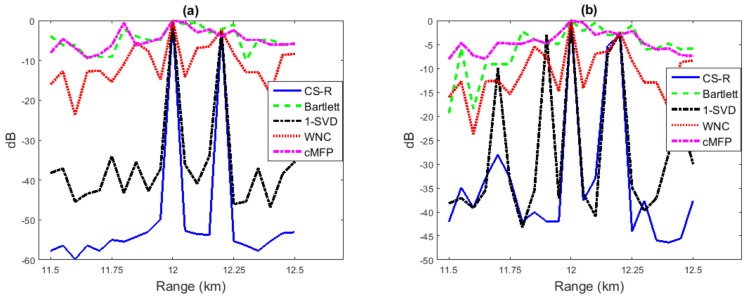
Two-source location estimation from 20 snapshots. Source 1 is located at a depth and range of 20 m and 12 km respectively with SNR = 30 dB, and Source 2 is located at a depth and range of 20 m and 12.2 km respectively with SNR = 8 dB. (**a**) Two-source localization in a known environment and (**b**) two-source localization with environmental mismatch.

**Figure 7 sensors-19-04522-f007:**
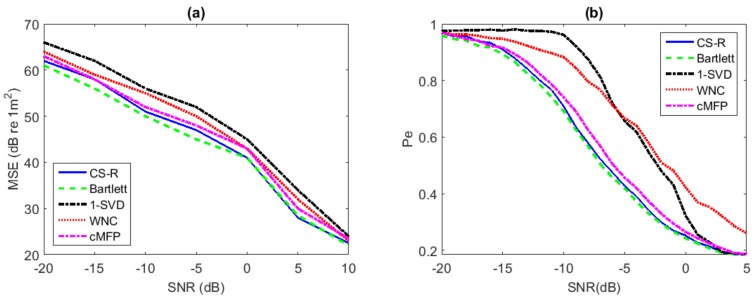
Performance analysis for source range estimation with environmental uncertainly: (**a**) MSE vs. SNR; (**b**) Pe vs. SNR.

**Figure 8 sensors-19-04522-f008:**
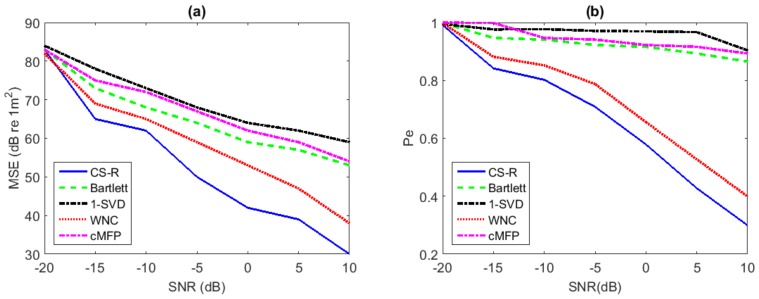
Performance analysis for source range estimation with environmental uncertainly while using two sources: (**a**) MSE vs. SNR; (**b**) Pe vs. SNR.

**Figure 9 sensors-19-04522-f009:**
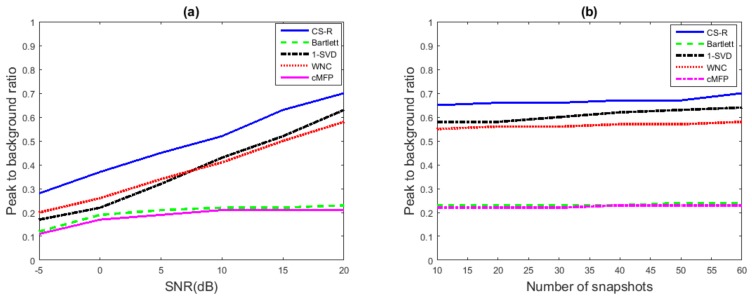
Peak to Background Ratio (PBR) comparison of CS-R with Bartlett, cMFP, 1-SVD, and WNC: (**a**) PBR vs. SNR; (**b**) PBR vs. number of snapshots.

**Figure 10 sensors-19-04522-f010:**
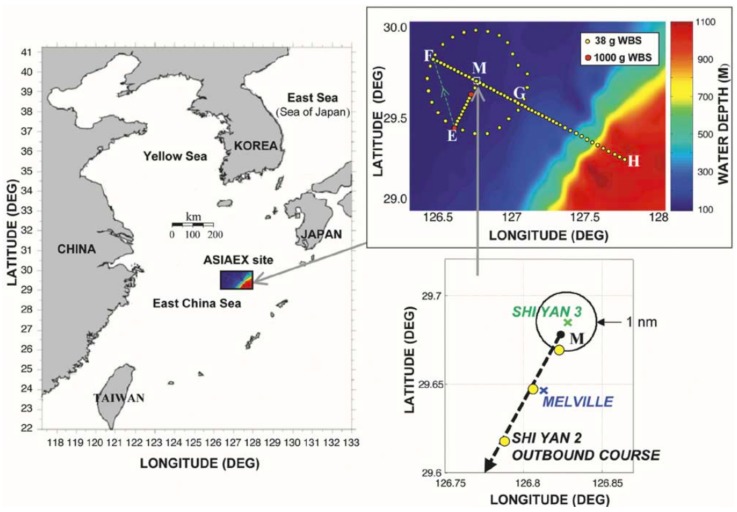
Location of the Asian Seas International Acoustics Experiments (ASIAEX) East China Sea (ECS) experimental site. Clockwise from left: large-scale view reducing to a small scale (lower right) showing the positions of the Shi Yan 3 (receiver) and R/V Melville near position M and the outbound course of the R/V Shi Yan 2 (source) [57].

**Figure 11 sensors-19-04522-f011:**
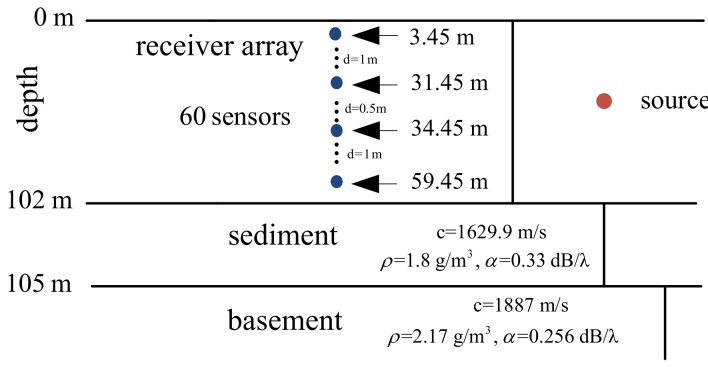
ASIAEX shallow water environment.

**Figure 12 sensors-19-04522-f012:**
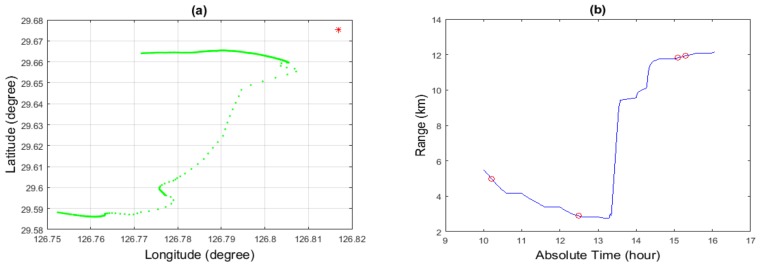
(**a**) Position of the Shi Yan 2 (red dot) and the movement of Shi Yan 3 (green line) from GPS coordinates; (**b**) change in distance between the source and receiver with respect to time. Red circle are showing the time and distance when the receiver is receiving the signal from the source.

**Figure 13 sensors-19-04522-f013:**
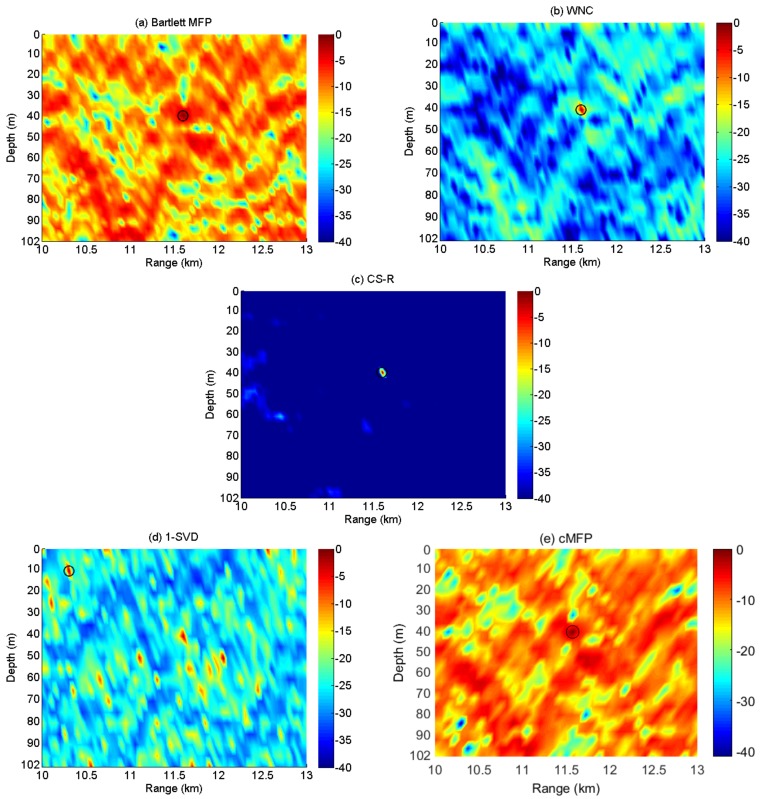
The ambiguity surface of (**a**) Bartlett; (**b**) WNC; (**c**) CS-R; (**d**) 1-SVD; (**e**) cMFP; using a vertical line array of 42 sensors and 30 snapshots in time. SNR = 15 dB; the source is at a range of about 11.6 km and a depth of about 40 m.

**Figure 14 sensors-19-04522-f014:**
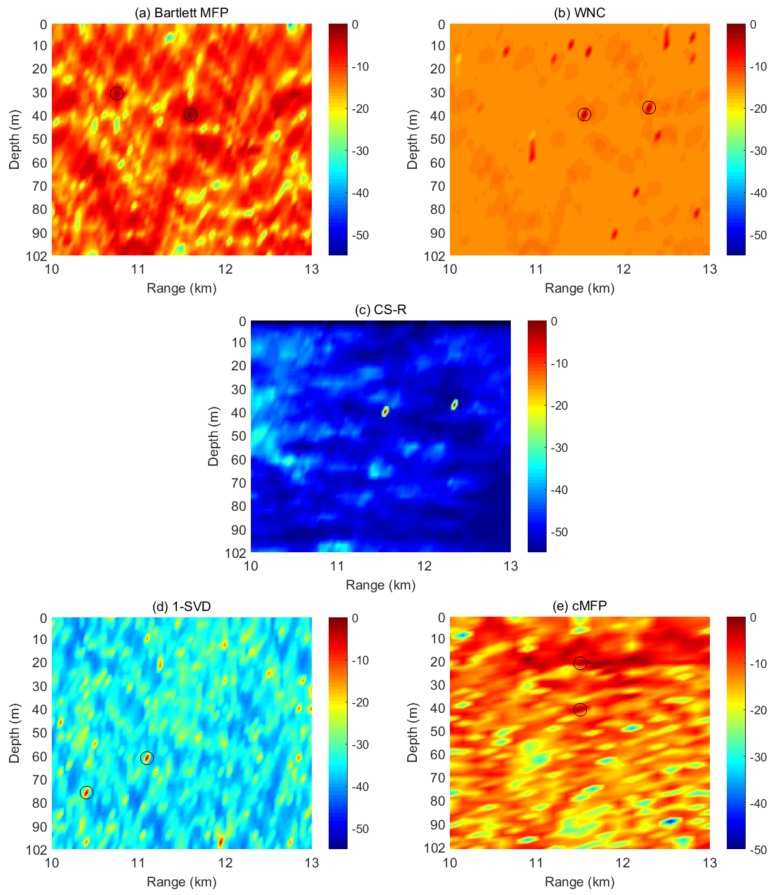
Two-source localization using experimental data. The first source is at a range of about 11.6 km and a depth of about 40 m, while the second source is located at a range of about 12.3 km and a depth of about 40 m. The ambiguity surfaces shown are for: (**a**) Bartlett; (**b**) WNC; (**c**) CS-R; (**d**) 1-SVD; (**e**) cMFP; using a vertical line array of 42 sensors and 30 snapshots in time. The SNR for the first and second sources are 15 dB, and 8 dB, respectively.

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
