# Peer review of "Passive Source Localization Using Compressive Sensing"

_sensors, 2019, doi:10.3390/s19204522_

Round 1

Reviewer 1 Report

This paper proposes an underwater passive source localization method using sparse-reconstruction-based MFP method. The proposed technique, CSR,  gives sharp peak and low side lobe level, ratio of measurements to sparsity (RMS) gains, and the provision of robust estimation.

The paper is well-written and gives a lot of background detail. The math and equations are presented well and clear. Extensive simulation results are shown that the CSR outperforms other techniques in various aspects. Some experimental results for East China Sea are shown.

One concern is the results from experimental results. it is not clear to me that if the authors have done the real hardware testing at the sea. or they use the data from ASIAEX to apply to the proposed method to get the results. The authors need to clarify it.

If some real testing is done using some H/W, the authors need to show the H/W and how to conduct the test.

If it is just using the data from ASIAEX, how exactly to use the data to generate your simulation results need to be explained.

Reviewer 2 Report

Please find enclosed my review.
